# Clinical Relevance of Gut Microbiota Alterations under the Influence of Selected Drugs—Updated Review

**DOI:** 10.3390/biomedicines11030952

**Published:** 2023-03-20

**Authors:** Honorata Mruk-Mazurkiewicz, Monika Kulaszyńska, Karolina Jakubczyk, Katarzyna Janda-Milczarek, Wiktoria Czarnecka, Ewa Rębacz-Maron, Sławomir Zacha, Jerzy Sieńko, Samir Zeair, Bartosz Dalewski, Wojciech Marlicz, Igor Łoniewski, Karolina Skonieczna-Żydecka

**Affiliations:** 1Department of Biochemical Science, Pomeranian Medical University in Szczecin, 71-460 Szczecin, Poland; 2Department of Human Nutrition and Metabolomics, Pomeranian Medical University in Szczecin, 71-460 Szczecin, Poland; 3Institute of Biology, Department of Ecology and Anthropology, University of Szczecin, 71-415 Szczecin, Poland; 4Department of Pediatric Orthopedics and Oncology of the Musculoskeletal System, Pomeranian Medical University in Szczecin, 71-252 Szczecin, Poland; 5Department of General and Gastroenterology Oncology Surgery, Pomeranian Medical University in Szczecin, 71-899 Szczecin, Poland; 6Institute of Physical Culture Sciences, University of Szczecin, 70-453 Szczecin, Poland; 7General and Transplant Surgery Ward with Sub-Departments of Pomeranian Regional Hospital in Szczecin, 71-455 Arkonska, Poland; 8Department of Dental Prosthetics, Pomeranian Medical University in Szczecin, 70-111 Szczecin, Poland; 9Department of Gastroenterology, Pomeranian Medical University in Szczecin, 71-455 Szczecin, Poland

**Keywords:** pharmacotherapy, drugs, phytochemicals, microbiota, interactions

## Abstract

As pharmacology and science progress, we discover new generations of medicines. This relationship is a response to the increasing demand for medicaments and is powered by progress in medicine and research about the respective entities. However, we have questions about the efficiency of pharmacotherapy in individual groups of patients. The effectiveness of therapy is controlled by many variables, such as genetic predisposition, age, sex and diet. Therefore, we must also pay attention to the microbiota, which fulfill a lot of functions in the human body. Drugs used in psychiatry, gastroenterology, diabetology and other fields of medicine have been demonstrated to possess much potential to change the composition and probably the function of the intestinal microbiota, which consequently creates long-term risks of developing chronic diseases. The article describes the amazing interactions between gut microbes and drugs currently used in healthcare.

## 1. Introduction

There is a need for constant discovery of newer generations of medicines. This process follows the response to the increasing demand for medicaments and is powered by progress in medicine and research about thee respective disease entities. However, the questions about the efficacy of pharmacotherapy in individual groups of patients remain. The effectiveness of therapy is controlled by many variables, such as a genetic predisposition, age, sex and diet. Therefore, one must also pay attention to the gastrointestinal microbiota, which support a lot of functions in the human body. 

The microbiota denote every microorganism in a particular habitat: bacteria, viruses and *eukaryota*. The most diverse and the most dense microenvironment for microorganisms is the gut. The gut microbiota may include around 1000 species of microorganisms, having their own metabolisms, and 150 times more genes than humans in total. The strength of the gut microbiota is correlated with the health status of a host; in the case of an imbalance between commensals and pathological microorganisms (referred to as dysbiosis), patients may have variety of symptoms or suffer from a certain disease [1]. The associations among drugs and the microbiota are complex, and for some pharmaceuticals, crucial for their efficacy. One of the most common processes for xenobiotics is hydrolysis of coupled or glycosidic compounds. This action is performed by beta-glucuronidase, beta-glucosidase and sulfatase, which all come from gut microbiota. Other than hydrolytic reactions, acetylation, deconjugation, modification of the thiazole ring, denitration and others, take place [2]. Beyond drug biotransformation, gut microbiota can regulate the activities of medication by direct changes to the host’s metabolism and synthesis of metabolites, which compete with receptors for relevant drugs. The microbiota’s metabolites can also regulate metabolism secretion of liver enzymes [3]. Overall, any change in the gut microbiota’s composition may result in a change in the pharmacokinetics’ pharmacodynamics and changes in the toxicity of the drugs that are metabolised in the digestive system. In both in vitro and in vivo studies, a correlation was observed regarding the pharmacokinetics and pharmacodynamics of over 30 drugs, including drugs with a narrow therapeutic range [4]. Nabumetone, a non-steroidal anti-inflammatory drug, is reduced by the microbiota and then metabolised in the liver to its active metabolite. However, incubation with one of the *E. coli* strains resulted in the formation of a reduced but inactive metabolite [5]. The clinically desired metabolites of deleobuvir (used in the treatment of hepatitis C) [6] and epacadostat (used in the treatment of cancer) [7] were created exclusively by the intestinal microbiota. Their incubation with subcellular populations of other tissues did not bring about efficacy.

In addition to external factors, it is drugs and their metabolites that have the ability to remodel the human microbiota. As evidenced in our last study [8], psychotropic drugs can affect the microbiota’s composition to a large extent. However, this is not always associated with a negative effect. For instance cyclophosphamide, which is an anti-cancer agent, causes the translocation of bacteria of the *Lactobacillus* and *Enterococcus* genera to secondary lymphoid organs, where they promote the differentiation of Th1 and Th11 lymphocytes, which are necessary for the anti-cancer effect of the drug [9]. Of a few groups of medicine, proton pump inhibitors (PPIs), statins, antibiotics, laxatives and metformin are the ones with the largest effects on microbial type [10].

The aim of this narrative review was to outline the interactions between the most commonly used drug groups (apart from antibiotics and psychotropic drugs, which are described elsewhere) and the microbiota, in order to understand the importance of these interactions for the therapeutic process, increase the effectiveness of the drugs and help reduce side-effects.

To comprehensively evaluate the issue, the literature searching process included: (1) Finding a set of studies aiming to look for the interactions between drugs and the gut’s microorganisms (in PubMed and Embase databases with the following search terms: microbiota, pharmacology, drugs and pharmacomicrobiomics). (2) Mannual searching of the reference list of reviews on similar topics. (3) Studying a selection of both animal and human studies, and discussion of mechanisms of action of drugs on the gut’s microbiome. The study selection was performed by the first and senior authors. This step took place over the course of two weeks. (4) Charting the data: main outcomes referring to drugs and gut microbiota. (5) Summarising data according to groups of drugs. 

## 2. Metformin

Metformin, a biguanide derivative, is the primary drug used in the treatment of type 2 diabetes. In addition, metformin is also administered to patients with insulin resistance, metabolic syndrome, polycystic ovary syndrome or lipodystrophy. Metformin exerts its action by inhibiting hepatic glucose production, increasing (insulin-dependent) glucose utilisation in skeletal muscle, reducing fatty acid oxidation and lowering triglyceride and LDL cholesterol levels. Unfortunately, in the course of therapy, patients often complain of gastrointestinal side-effects, such as a lack of appetite, nausea, abdominal pain, bloating, diarrhoea, constipation, anaemia and lactic acidosis. Metformin accumulates in the gastrointestinal mucosa, causing local inflammation, which is responsible for the symptoms listed above. In addition, metformin interferes with the normal absorption of bile salts, which contributes to the exacerbation of discomfort and causes additional symptoms, such as loose stools [11]. A recent meta-analysis showed that using metformin, compared to other antidiabetic drugs, significantly elevates the risk of occurrence of gastrointestinal symptoms—among others, abdominal pain and diarrhoea—and there are some advantages to using an extended-release formulation [12]. The observed side-effects may be related to the effect of the drug on the microbiota [13]. A Colombian study showed that the use of metformin in patients with type 2 diabetes increased the abundance of *Akkermansia muciniphila* and several short-chain fatty acid producing bacteria, such as *Butyrivibrio, Bifidobacterium bifidum* and *Megasphaera.* Importantly, *A. muciniphila* has multiple beneficial effects on metabolism [14]. One of its cell wall proteins, namely, Amuc1100, was experimentally proved to regulate carbohydrate metabolism. Recently, a postbiotic intervention was successfully used on patients suffering from diabetes and obesity, improving their insulin response [15], anthropometric indices and hepatic wenzymes [16]. In addition, it was demonstrated that the higher the abundance of *A. muciniphila*, the more mucin-producing goblet cells, which increases during metformin treatment [17]. Forslund et al., in their analysis of 784 human gut metagenomes, proved that patients treated with metformin showed a significant reduction in *Intestinibacter* and an increase in *Escherichia* [18]. The former of these genera has been described as predisposing to susceptibility to enterotoxigenic *Escherichia coli* F4 [19]. In contrast, analysis of the functional potential of the gut’s metagenome in the cited study demonstrated that the indirect effects of metformin treatment associated with reduced intestinal lipid absorption and bacterial lipopolysaccharide-triggered inflammation can boost the abundance of *Escherichia coli* [20]. These changes can lead, among other things, to gastrointestinal disorders, as illustrated in the diagram below (Figure 1).

Similar observations were performed in a group of 27 healthy men given metformin for 18 weeks. The abundances of 11 bacterial genera changed during the intervention but returned to baseline levels after metformin was discontinued. Reduced numbers of *Intestinibacter* spp. and *Clostridium* spp. were observed, along with higher numbers of *Escherichia* and *Shigella* spp. [13]. Karlsson et al. conducted a cohort study in a group of 145 European women and showed that the microbiomes of metformin users contained higher abundances of 103 bacterial species. These were mainly bacteria from the family *Enterobacteriaceae* (*Escherichia coli*, *Shigella*, *Klebsiella*, *Citrobacter*, *Enterobacter cloacae*, *Salmonella enterica*). In contrast, reduced abundance was observed for 18 species, including *Clostridium bartlettii*, *Mobiluncus mulieris, Peptoniphilus duerdenii* and *Eubacterium siraeum* [21]. De la Cuesta-Zuluaga et al. showed that metformin treatment results in higher abundances of *Prevotella* and *Megasphaera* and lower abundances of *Oscillospira, Barnesiellaceae* and *Clostridiaceae* [22]. A multicentre, double-blind, randomised trial involving 160 children with obesity was conducted in Germany and Spain. It was noted that in children receiving placebo, the abundances of *Actinobacteria* and *Bacillus* were higher than in those treated with metformin [23]. Similar observations were made by the research team of Xiang et al. [24]. This finding is important because the abundances of these types of microorganisms are known to be elevated in obese people with metabolic disorders [25,26]. Elbere et al. observed a reduction in the diversity of the gut microbiota 24 h after metformin administration, and a positive association between the severity of gastrointestinal symptoms and the abundance of *Escherichia-Shigella* spp. After one week of treatment, the abundances of *Peptostreptococcaceae* and *Clostridiaceae* decreased [27]. In patients with type 2 diabetes, metformin caused increased abundance of *Escherichia* and *Bifidobacterium*. Faecal transplantation in germ-free mice resulted in improved glucose tolerance [24]. Metformin is also of great interest in the management of metabolic disorders in patients treated with second-generation antipsychotics [28], in the formation of which, dysbiosis plays a significant role [29]. 

## 3. Non-Steroidal Anti-Inflammatory Drugs

Non-steroidal anti-inflammatory drugs (NSAIDs) are among the most commonly used pharmaceuticals for relieving pain, inflammation, and fever. They work by blocking the production of prostaglandins through the inhibition of cyclooxygenases involved in their synthesis. The use of these drugs carries risks of a number of adverse effects—most commonly heartburn, oesophagitis and gastritis, indigestion, dyspepsia, epigastric pain and the formation of gastrointestinal ulcers. Recent scientific reports also highlight their roles in modifying the function and composition of the gut microbiome. In a US study of 155 adults, Rogers and Aronoff attempted to demonstrate a link between the use of non-steroidal anti-inflammatory drugs and the composition of the gut microbiota [30]. Significant alterations were observed in the composition of the gut microbiota in people taking anti-inflammatory drugs. Importantly, it appears that the composition was influenced by the type and not the dose of the medication used. The use of acetylsalicylic acid resulted in higher abundances of *Prevotella*, *Bacteroides* and *Barnesiella* species and the family *Ruminococcaceae*. In addition, the gut microbiota of people using celecoxib were similar to those of ibuprofen users—containing higher abundances of *Acidaminococcaceae* and *Enterobacteriaceae*. Higher prevalences of bacteria from the families *Propionibacteriaceae*, *Pseudomonadaceae*, *Puniceicoccaceae* and *Rikenellaceae* were observed in ibuprofen users compared to controls and naproxen users. *Bacteroides* species and bacteria from the family *Ruminococcaceae* were more abundant in users of NSAIDs in combination with antidepressants and laxatives than in users of NSAIDs alone. Furthermore, combination therapy with NSAIDs and proton pump inhibitors was associated with the presence of *Bacteroides* and *Erysiplotrichaceae* species, which were not observed in individuals using NSAIDs alone [30]. Makivuokko et al. showed that the use of NSAIDs in the elderly resulted in reductions in *Dialister*, *Coprobacillus* and *Collinsella* amounts compared to young adults and greater abundances of *Lactobacillus*, *Collinsella aerofaciens* and *Roseburia* in individuals of the same age not using NSAIDs [31]. Bokulich et al. found no effects on the microbiota after celecoxib treatment at 2 × 200 mg daily for 10 days [32]. Edogawa et al. administered indomethacin at 75 mg 2 × daily for 5 days and collected stool samples from 11 men and 12 women. In the material studied, decreases in the amounts of *Firmicutes* and *Ruminococcus* were seen in women, and increases in the same bacterial populations were observed in men [33]. In research reports, it is emphasised that NSAIDs do not alter the richness or diversity of gut microbiota. In the study by Prizment et al., no such alterations were observed after 6-week treatment with aspirin [34]. These conclusions are also supported by other research [35]. NSAID-induced gut-microbiota alterations are strictly linked to the type of a molecule used. Celecoxib and ibuprofen usage resulted in increases in *Enterococcaceae* and *Erysipelotrichaceae* abundances. Researchers have highlighted an important bidirectional relationship between the gut microbiome and NSAID use. This has already been demonstrated in several rat studies. Roberto and Asano showed that GF rats were resistant to intestinal lesions induced by indomethacin use, but these lesions did occur after *E. coli* infection. In another study, amoxicillin-induced dysbiosis was shown to slow the effect of acetylsalicylic acid [36]. Antibiotic-induced dysbiosis has also been shown to alter the pharmacokinetics and pharmacodynamics of indomethacin [37].

## 4. Proton Pump Inhibitors (PPIs)

PPIs are a widely used group of drugs that, due to their mechanism of action—increasing gastric pH—significantly influence the ecology of the gastrointestinal tract. Their action can have significant clinical implications and result in adverse effects, including diarrhoea, nausea and/or vomiting, abdominal pain and headaches [38]. The use of proton pump inhibitors significantly affects the microbial habitats in the oral cavity, the oesophagus and the small and large intestine. The relative stability of the gut microbiome of the organs listed above is disrupted during the use of PPIs [39]. The oral microbiota is mainly composed of *Firmicutes* and *Bacteroidetes*; and also *Actinobacteria*, *Proteobacteria* and *Fusobacteria*. The current state of knowledge about the effect of antacids on the composition of the oral microbiome is quite poor. In one study, it was shown that 4-week administration of esomeprazole in patients was associated with increases in the bacterial populations of *Fusobacterium* and *Leptotrichia* in the periodontal pocket, and increases in *Neisseria* and *Veillonella* in saliva [39]. Downstream, along the gastrointestinal tract, the microbiome also remains stable. On the basis of differences in microbial composition, two types of microbiota can be identified. Type I, composed predominantly of Gram-positive bacteria, is characteristic of healthy people. In contrast, type II, associated with, e.g., Barrett’s oesophagus, consists mainly of Gram-negative bacteria. The switch from a type I to type II oesophageal microbiome is associated with chronic and recurrent mucosal inflammation and dysbiosis [39]. PPI treatment results in an increase in *Firmicutes* and decreases in *Bacteroidetes* and *Proteobacteria* [40]. The use of proton pump inhibitors has an adverse effect on both the composition of the gastric microbiota and its functions, i.e., delayed gastric emptying, reduced viscosity of gastric mucus and increased translocation of bacteria [41]. During therapy, marked increases have been noted in bacterial populations such as *Streptococcaceae*, *Prevotellaceae* and *Campylobacteraceae*. Furthermore, the effect on gastric pH appears to be a key element influencing both gastric dysbiosis and the development of gastric cancer. However, the potential impact of PPI use on the development of gastric cancer in long-term users of proton pump inhibitors remains the subject of ongoing academic debate. Small intestinal bacterial overgrowth (SIBO) is one of the main effects of long-term PPI treatment, which is characterised by an increase in aerophilic microorganisms. In addition, treatment-induced dysbiosis is a risk factor for hepatic encephalopathy and spontaneous peritonitis in patients with cirrhosis [42]. It is also noteworthy that there are reports highlighting increased damage to the intestinal mucosa by the use of PPIs in combination with NSAIDs [43,44]. The treatment promotes the development of intestinal dysbiosis, which promotes mucosal damage in the intestines [43]. The use of proton pump inhibitors contributes to an increased risk of intestinal infections, particularly involving *C. difficile*, *Salmonella*, *Campylobacter* and *E. coli* [45]. Colonisation by the above-mentioned microorganisms alters the composition of the intestinal microbiota, and consequently, the implemented treatment significantly aggravates dysbiosis and dysfunction of the gut–brain axis, leading to the development of, among other things, irritable bowel syndrome [46]. In addition to functional disorders, the use of proton pump inhibitors contributes to hypergastrinemia, which is thought to be the cause of excessive proliferation of colonic epithelial cells, leading to adenoma development [47]. In their study, Davis et al. [48] sought to find an association between the abundance of *Akkermansia muciniphila* and obesity, with consideration given to body composition, metabolism and lifestyle factors. The study involved 158 men who had their faecal samples examined by 16S rRNA intestinal metagenome sequencing and underwent thorough interviews to collect lifestyle information. The abundance of the studied bacteria was estimated from total sequence reads, and the degree of obesity was quantified by the fat mass Index (FMI). It was noted that the most common medications in the study group were proton pump inhibitors, and their use was associated with both high FMI and decreased abundance of *A. muciniphila*. The relative abundance of *A. muciniphila* was inversely associated with high FMI, independent of PPI use. The researchers emphasise that the relationship between obesity, PPI use and gut-microbiota composition requires further investigation. Imhann et al. [49] conducted an analysis of a total of 1815 participants taking PPIs. The authors observed a reduction in microbial diversity according to the Shannon index and significant alterations in 20% of the bacterial taxa present. There were increases in *Enterococcus*, *Streptococcus*, *Staphylococcus* and the possibly pathogenic species *Escherichia coli*. Bacteria from the order *Actinomycetales*, families *Streptococcaceae* and *Micrococcaceae*, genus *Rothia* and species *Lactobacillus salivarius* were also observed in increased numbers. On the other hand, none of individual bacterial species were present in significantly reduced numbers. Jackson et al. analysed the association between the gut microbiome and use of proton pump inhibitors in a cohort of 1827 healthy twins in the United Kingdom. The treatment has been shown to reduce the diversity of the gut microbiota [50].

## 5. Statins

Statins, or hydroxymethylglutaryl-CoA (HMG-CoA) reductase inhibitors, act via inhibiting sterol biosynthesis. Statins efficiently lower cholesterol and are therefore used in the prevention of cardiovascular disease. To date, seven statin molecules have been discovered, differing in terms of bioavailability, lipo/hydrophilicity, cytochrome P-450-mediated metabolism and cellular transport mechanisms [51]. Notwithstanding their clinical success, there are also some side-effects to the use of statins, including increased incidences of diabetes and cataracts, and frequent muscular side-effects. 

Statins also can alter the composition of the gut microbiota [52]. Hu et al. found that such medication can restore gut microbiota homeostasis, which can consequently improve outcomes in the case of acute coronary syndrome (ACS) [53]. There is evidence that specific changes in the gut microbiota and microbial metabolites influence the progression of coronary artery disease, and that a history of long-term statin treatment is correlated with more favourable outcomes in patients with ACS. In the study, 16S rRNA sequencing and serum metabolomic analysis were performed in 36 patients with ACS who had been treated with statins on a long-term basis, 67 ACS patients who had no history of statin treatment and 30 healthy volunteers. Metagenomic functional prediction of the bacteria was performed with the advent of a PICRUSt2 tool. Statins were found to act positively in ACS patients in whom the drug reduced the abundance of potentially pathogenic *Parabacteroides merdae*, and simultaneously increased *Bifidobacterium longum* subsp. *Longum*, *Anaerostipes hadrus* and *Ruminococcus obeum*, which are of a beneficial nature. Statin-related taxa abundance were linked to fatty acid and isoprenoid biosynthesis pathways. In addition, the drug affected the prognosis positively in ACS patients. The multiomics approach enabled the researchers to discover that drug-induced alterations in microbiota were associated with disease severity or outcomes either directly or indirectly via fatty acids and prenol lipids’ metabolism. These findings provide new insights into the heterogeneous roles of statins in ACS patients through metabolic interactions of the host’s gut microbiota [53]. Statins exhibit synergistic activity with antibiotics, including antimicrobial effects and the ability to stimulate the host’s immune system, which makes them useful as adjunctive drugs in the case of antimicrobial resistance (AMR) [54]. The supportive role of statins in antibiotic therapy is extremely valuable; unfortunately, their current widespread use in cardiovascular protection may result in the development of a kind of resistance to their antimicrobial action, which will exacerbate AMR instead of alleviating it. Ko et al. [54] investigated the antimicrobial activity of statins, demonstrating that simvastatin generally exerted the greatest antimicrobial effect against Gram-positive bacteria compared to atorvastatin, rosuvastatin and fluvastatin. Against Gram-negative bacteria, atorvastatin generally produced similar or slightly better results compared to simvastatin, but both were more potent than rosuvastatin and fluvastatin. The antibacterial effect of statins is based on the ability to both bind and degrade structures found in bacteria, such as surface proteins, lipoteichoic acid and lipopolysaccharides. The effect of statins has been described in patients taking a therapeutic dose, so it is likely that the minimum bactericidal concentration is lower than the concentration of the drug in plasma [54].

Kim et al. investigated gut-microbiota modulation by statins (atorvastatin and rosuvastatin) in a mouse model of obesity induced by a high-fat diet and described the relationship between gut-microbiota composition and the immune response. Atorvastatin and rosuvastatin significantly increased the abundances of the genera *Bacteroides*, *Butyricimonas* and *Mucispirillum*. Furthermore, the abundances of these bacterial genera were correlated with the inflammatory response, including levels of IL-1β and TGFβ1 in the ileum. Oral microbiota transplantation using faecal material collected from rosuvastatin-treated mice improved hyperglycaemia. These findings suggest that modulation of the gut microbiota by statins plays an important role in the therapeutic activity of these drugs [55]. To investigate the effect of simvastatin on the gut microbiota of hyperlipidaemic rats, the molecular characteristics of the gut microbiota and the potential functions of the genes involved in downstream metabolic pathways were analysed. The results revealed that simvastatin (SIM) treatment can reduce the gut’s microbial diversity and induce marked remodelling of the faecal bacterial community’s composition [56]. This was the first study to establish a profound and comprehensive relationship between SIM-induced alterations in the gut microbiota and changes in metabolic pathways by the gut microbiota. Its findings suggest that the gut microbiota may contribute to the hypolipidemic efficacy of SIM in the progression of hyperlipidaemia [56].

As evidenced above, statins might increase the risk of T2D, but on the other hand, a faecal transplant form rosuvastatin treated mice improves glucose metabolism. Indeed, the mechanism of diabetes development during statin therapy is not fully known. It might be linked to the inhibition of calcium channels in the pancreas, leading directly to a decrease in insulin secretion. Additionally, the occurrence of polymorphisms in several genes, causing impaired proinsulin to insulin conversion, might be the cause. Another probable mechanism of the development of diabetes after statin therapy is related to the fact that statins affect the size of lipoproteins. However, the experimental model provided information about the beneficial effects of rosuvastatin and atorvastatin, which modelled the sizes of the lipoproteins, and subsequently made them protective in diabetes [57,58]. 

The composition of the gut microbiota was found to be an independent factor affecting one’s response to therapy. Wang et al. examined the impact of the gut microbiome’s architecture on the response to statins in the case of coronary artery disease (CAD). A group of 836 CAD patients were divided into two groups according to their responses to statins (good and poor response groups) to compare their gut-microbiota compositions. The analysis showed no significant differences in the microbiome (i.e., the genetic pool of the microorganisms found) between groups. However, significant alterations in the overall compositions of the gut microbiota of participants were noted. Decreases in the lipid profile improved *Akkermansia muciniphila and Lactobacillus,* and significant increases in *Holdemanella* and *Faecalibacterium*, were negatively linked to drug response. These results suggest steering the gut-microbiome composition (via, for instance, drugs) might be effective for lipid metabolism in CAD [59]. As elegantly demonstrated by Zhao et al., fluvastatin (FLU2) improved the growth of *Escherichia/Shigella*, *Ruminococcaceae UCG 014* and *Sutterella*. The function of the gut microbiota, expressed as the synthesis of short-chain fatty acids (SCFAs), was also significantly decreased compared to the use of other statins. In contrast, the usage of rosuvastatin (ROS), simvastatin (SIM) and atorvastatin (ATO) did not have a significant impact on gut microbiome. The composition of the gut microbiota, in turn, influences statin metabolism in the body [60]. Nolan et al. investigated the effect of rosuvastatin (RSV) on the composition of the gastrointestinal microbiota in mice. RSV significantly affected the microbial architecture of the cecum and large intestine. Diminished diversity and lower abundances of some physiologically relevant bacterial groups are typical for the cecum. RSV had a significant impact on the metabolism of bile acids and the expression of inflammatory biomarkers, affecting the gut barrier itself. One study suggests that RSV impacts the gut microbiota in mice, affecting local gene expression profiles [61].

## 6. Laxatives

The term laxatives refers to plant products, chemical compounds or medicines that increase bowel movements and loosen the stool, thereby stimulating defecation. They are taken orally or in the form of suppositories, usually to relieve constipation. Functional constipation is a common, debilitating gastrointestinal disorder, which remains difficult to treat. Combination therapy combining multiple interventions produces the best results. Laxatives are widely used to provide temporary relief from constipation, but their overall efficacy is unsatisfactory. Faecal microbiota transplantation (FMT), which appears to be an aetiological treatment, is an increasingly studied and used intervention [62]. Chu et al. in a randomised, placebo-controlled intervention study examined the efficacy of prebiotic UG1601 in patients suffering from functional constipation. Adults with a stool frequency of less than three times a week were randomised to receive either the prebiotic or a placebo. The intervention lasted four weeks. All participants provided stool and blood samples at baseline and at the end of the intervention. The researchers evaluated gastrointestinal symptoms and stool frequency, and serum concentrations of endotoxemia markers and faecal short-chain fatty acids (SCFAs). Additionally, the relative abundances of SCFA-producing bacteria and the composition of the gut microbiota were assessed in both responders and non-responders to prebiotics. As for the differences in gastrointestinal symptoms, there were no significant ones between the groups, but responders reported improvements in constipation. In this group, the abundances of Firmicutes and the family *Lachnospiraceae* (phylum Firmicutes, class *Clostridia*) decreased. At the molecular level, prebiotic intake diminished the level of the expression of serum cluster of differentiation (CD) 14 and lipopolysaccharide (LPS). In contrast, their gut microbiota were enriched in the main butyrate producer, *Roseburia hominis*. Furthermore, the abundances of the phylum *Firmicutes*, class *Clostridia* and order *Clostridiales* were inversely correlated with several faecal SCFAs. Based on the results, the authors concluded that changes in the composition of the gut microbiota, including a decrease in *Firmicutes* and an increase in butyrate-producing bacteria, following supplementation with the prebiotic UG1601, may help alleviate gastrointestinal symptoms and endotoxemia [63]. A number of randomised controlled trials of functional constipation treatment by FMT together with laxatives and laxatives alone in adults were compared in a systematic review and meta-analysis. Screening, data extraction and bias assessment were carried out independently by two reviewers. A total of 1400 records were identified, of which five met the criteria (409 patients). On the basis of the meta-analysis, Fang et al. confirmed that treatment combining laxatives with FMT significantly improved the parameters studied and improved quality of life scores in patients suffering from constipation compared to laxatives alone [64]. Takayama et al. in a murine study monitored the gut microbiota before and after the administration of laxatives in mice fed a high-carbohydrate, a high-fat or a high-fibre diet. Twenty mice per diet were divided into four groups to evaluate the laxative effects of four laxative preparations—Daiokanzoto (Da-Huang-Gan-Cao-Tang, DKT), sennoside A (SA), SA plus rhein 8-O-β-D-glucopyranoside (SA+RG) and SA plus liquiritin (SA+LQ). Changes in the gut microbiota were monitored by next-generation sequencing of 16S rRNA gene amplicons. In mice fed a high-carbohydrate and a high-fat diet, DKT had significantly greater purgative activity than SA alone, and RG contributed to this activity. Administration of DKT and SA+RG increased the content of *Enterobacteriaceae*, which was associated with an increased purgative activity. In contrast, the laxative properties of DKT were significantly suppressed by a high-fibre diet. The results suggest that diet-induced alterations in the composition of the gut microbiota determine the effects of DKT and other anthranoid laxatives [65]. Lactulose is a laxative predominantly used in cases of constipation. Zhang et al. [66] administered lactulose in a mouse model with loperamide-induced constipation. The intervention lasted 4 weeks and resulted in an increase in intestinal motility and simultaneously inhibited the inflammatory response and improved gut barrier integrity and the metabolism of water and salt in the gut. These therapeutic effects were attributed to significant changes in gut-microbiota functions, especially increases in bile acids, short-chain fatty acids and tryptophan catabolites’ production [66]. Tomkovich et al. conducted an intervention study in which C57BL/6 mice were treated with 15% polyethylene glycol (PEG) in drinking water for 1 or 5 days and then exposed to *C. difficile* 630 infection. The researchers also investigated whether osmotic laxatives might somehow affect the risk of *Clostridioides difficile* infection in a mouse model of *C. difficile* challenge. It was found that the drugs might impair resistance to *C. difficile* colonisation and prevent clearance in mice infected with this genus [67]. Matsuura et al. tried to identify the appropriate treatment for elderly patients suffering from chronic constipation and investigate its effect on intestinal immunity. Patients with defecating difficulties were randomly divided into two groups. Group A was given laxatives only, and group B was given laxatives in combination with probiotics as an intervention. The two groups were compared for the degree of improvement in constipation and the effects on the intestinal environment. After the administration of probiotics in combination with a laxative, patients experienced a significant improvement. It was also observed that changes in the composition of the gut microbiota before and after the intervention were associated with a significant improvement in the condition [68].

## 7. Phytochemicals

Medicinal plants have been used for thousands of years in many ailments for prevention, treatment and supportive care, being readily available, cheap and effective. Contemporary studies, both in vitro and in vivo, confirm their positive effects on the functioning of the human body. In addition to their many properties, including anti-inflammatory, analgesic, antipyretic and antihistamine effects, etc., botanicals affect the qualitative and quantitative composition of the gastrointestinal microbiota [69]. It turns out that the composition of the gut microbiota is closely linked to specific disease entities and being susceptible to modulation by diet and medication, including herbal remedies. In recent years, researchers have been investigating the effects of medicinal plants on the gut microbiome in selected disease entities. Researchers focus in particular on diseases of major epidemiological importance, such as cardiovascular conditions (e.g., ischaemic heart disease), diabetes and mental health [70,71,72,73]. Recent years have shown that herbs can also support COVID-19 therapy via boosting the immune system. Herbs and prebiotics positively affect gut-microbiota homeostasis and prevent a patient from secondary bacterial infections. Herbal-based medicines might, for instance, inhibit SARS-CoV-2 bloom with the advent of a special protease activity [74]. The effects mentioned above can be attributed to phytochemicals (phytonutrients), i.e., chemical compounds of various structures produced by plants. They do not serve essential functions and play no role in the primary metabolism of plants, being regarded as secondary metabolites. These compounds are involved in the ecological interactions between the plant and its environment, e.g., by providing protection against UV radiation, exerting bactericidal, fungicidal and insecticidal effects, and thus protecting the plant from unfavourable environmental conditions [75]. Due to the contents of various phytochemicals with different mechanisms of action, herbs can modulate the abundances of individual bacterial species, inhibiting bacterial translocation and damage to the intestinal barrier. Interactions between the gut microbiota and herbal medicines occur mainly via two pathways. Firstly, the gut microbiota “digests” herbal ingredients into bioavailable active small molecules that enter the body and induce physiological changes. Secondly, herbal medicines regulate the quantitative and qualitative composition of the gut microbiota and its secretions, and induce physiological changes [76]. The intestinal microflora has been shown to be altered by, among others, carotenoids, polyphenols, phytosterols/phytostanols, lignans, alkaloids, glucosinolates and terpenes [77,78,79,80]. Examples of the effects of plant products on the gut microbiome are shown in Table 1. 

It has also been shown that nuts, which are a rich source of monounsaturated fatty acids, tocopherols (almonds and hazelnuts), phytosterols and carotenoids (pistachios) [81], play a role in remodelling of the gut microbiota (Table 2).

## 8. Conclusions and Future Perspectives

In recent years, we have been witnessing significant progress in medicine, which resulted in the availability of new drugs and better treatment options in communicable and non-communicable diseases. Nevertheless, special attention should be paid to a variety of factors, which significantly affect the effectiveness of the treatment protocols. One of such factors is the gut microbiota, whose composition and function largely reflect the condition of the organism. The knowledge of the involvement of the intestinal microbiome in drug metabolism creates a basis for studying the novel relationships between the intestinal microbiota and pharmacotherapy. This has already been viewed as a new direction in medicine and clinical practice. As we have demonstrated, the effects of individual drugs and phytochemicals on both the structure and function of the gut microbiome (Table 3) should be taken into account in research and clinical practice. Consequently, when administering a particular drug to a patient, special attention should be paid to their potential effects on the gut microbiome, which can translate not only into the metabolism of the substances used, but most importantly and vice versa, into the well-being of the patient. Therefore, more studies are critically needed to improve the importance of this area of science and patient care.

## Figures and Tables

**Figure 1 biomedicines-11-00952-f001:**
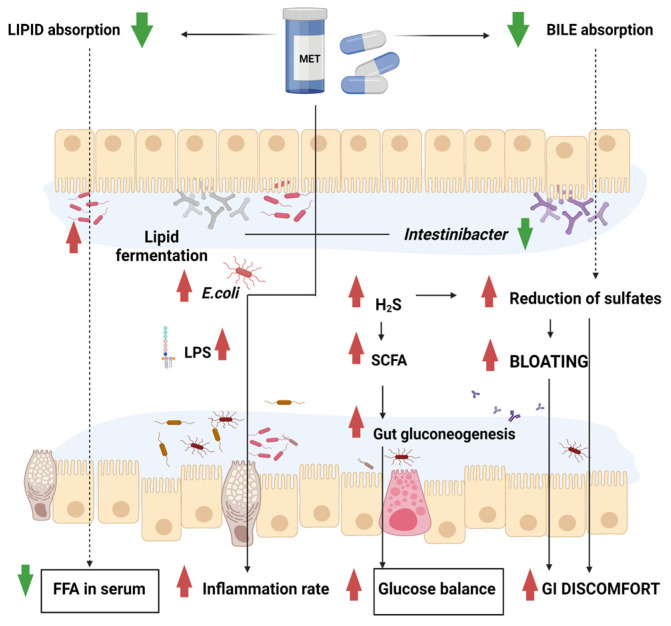
Metformin-induced changes in the intestinal barrier. FFA—free fatty acids, LPS—lipopolysaccharide, SCFA—short-chain fatty acids. Metformin increases the glucose balance and decreases lipid and bile absorption in the gut, thereby diminishing the level of FFA in serum. On the basis of taxonomic analyses, it turned out that the drug decreased the content of *Intestinibacter* but simultaneously enhanced the bloom of *E. coli*. At a functional level, enhanced butyrate and propionate production in metformin-treated individuals occurred, along with the enrichment of gas metabolism genes, all leading to elevated GI discomfort.

**Table 1 biomedicines-11-00952-t001:** Examples of the effects of phytochemicals on the gut microbiome [75,78].

Phytochemicals	Source	Part of Plant	Research Model	Effect on Bacterial Counts	
				Increase	Decrease
phenolic acids, stilbenes, flavonols, dihydroflavonols, anthocyanins	grapevine *(Vitis vinifera*)	fruits	Human	*Enterococcus* spp., *Prevotella* spp., *Bifidobacterium* spp., *Bacteroides uniformis*, *Eggerthella lenta*, *Blautia coccoides*, *Eubacterium rectale*	
chlorogenic acid	coffee *(Coffea arabica*)	green beans	animal (mice)	*Bacteroides*, *Bifidobacterium*, *Lactobacillus*, *Ruminococcaceae*	*Desulfovibrionaceae*, *Lachnospiraceae*, *Erysipelotrichaceae*
caffeine	coffee (*Coffea arabica*)	roasted beans	human	*Bifidobacterium* spp. *Faecalibacterium, Roseburia*	*Erysipelatoclostridium*
phenolic compounds	garlic *(Allium sativum*)	whole plant	in vitro		*Staphylloccocus epidermidis*, *Klebsiella pneumoniae*
allicin	garlic (*Allium sativum*)	bulbs	animal (mice)	*Akkermansia* spp.	
Tannins	moringa *(Moringa oleifera*)	leaves	animal (mice)	*Bacteroides*	*Clostridium leptum*
anthocyanidins, flavonoids	pomegranate *(Punica granatum*)	fruits	animal (mice)	*Lactobacillus* spp., *Bifidobacterium*	*Escherichia coli*, *Salmonella* spp.
Anthocyanidins	shrubby blackberry *(Rubus fruticosus*), raspberry *(Rubus occidentalis*)	fruits	in vitro		*Helicobacter pylori, Staphylococcus epidermis*, *Klebsiella pneumoniae*, *Escherichia coli*, *Salmonella* spp.
	Garden snapdragon (*Antirrhinum majus*)	flowers	animal (mice)	*Bifidobacterium* spp., *Lactobacillus* spp.	
Catechins, flavanols	green tea *(Camelia sinensis*)	leaves	human	*Lactobacillus* spp.	*Streptococcus mutans*, *Shigella*, *Vibrio cholera*, *Clostridium histolyticum*, *Clostridium coccoides*, *Eubacterium rectale*
procyanidins, catechins, epicatechins	apple *(Pyrus malus*)	fruits	animal (mice)	*Bifidobacterium* spp.	
Coumarins	cashew *(Anacardium occidentale*)	leaves	*in vitro*		*Staphylococcus aureus*
sulforaphane	brassicaceous vegetables (e.g., radish, cabbages, cauliflower, broccoli)	edible parts	animal (mice)	*Bacteroides fragilis*, *Clostridium* cluster	
polyphenols	black currant *(Ribes nigrum*)	leaves and fruits	human	*Lactobacilli, Bifidobacteria*	*Clostridium* spp., *Bacteroides* spp.
Quercetin, rutin, (or buckwheat supplementation)	buckwheat (*Fagopyrum esculentum*)	seeds	human	*Eubacterium ramulus*	
flavanol	cocoa (*Theobroma cacao*)	fruits	human	*Bifidobacterium* spp., *Lactobacillus* spp.	*Clostridium* spp.
Polyphenols	red wine	grapes		*Enterococcus, Prevotella*, *Bacteroides*, *Bifidobacterium*, *Bacteroides uniformis*, *Eggerthella lenta*, *Blautia coccoides*, *Eubacterium rectale*	
β-carotene	isolated pure chemical compound		human	*Firmicutes*	*Bacteroides*
lycopene	isolated pure chemical compound		human	*Bifidobacterium adolescentis*, *Bifidobacterium longum*	*Bacillus subtilis*
terpenes, aroma active compounds	rosemary *(Rosmarinus officinalis*)	herbaceous	animal (rats)	*Bifidobacterium* spp., *Blautia coccoides*, *Bacteroides* spp.	*Lactobacillus*, *Leuconostoc*, *Pediococcus*, *Clostridium* spp.

**Table 2 biomedicines-11-00952-t002:** Examples of the effects of nut ingredients on microbiota changes [81].

Source	Research Model	Effect on Bacterial Counts	
		Increase	Decrease
defatted almonds	in vitro	*Bifidobacteria*, *Eubacterium rectale*	
almonds (raw and roasted)	in vitro	*Lactobacillus acidophilus*, *Bifidobacterium breve*	*Escherichia coli*
	animal (rats)	*Lactobacillus* spp., *Bifidobacterium* ssp.	*Enterococcus* spp. *Escherichia coli*
chestnut	in vitro	*Lactobacillus paracasei GG*, *Lactobacillus rhamnosus*, *Lactobacillus casei*	*Streptococcus macedonicus Streptococcus thermophilus*
walnuts	animal (rats)	*Lactobacillus, Ruminococcaceae, Roseburia*	*Bacteroides Anaerotruncus Alphaproteobacteria*
	human	butyrate-producing bacteria, *Ruminococcaceae*, *Bifidobacteria*	*Clostridium* spp.
roasted almond, almond skin	human	*Lactobacillus* spp., *Bifidobacterium* spp.	*Clostridium perfringens*
whole almonds, roasted almonds, roasted chopped almonds, almond butter	human		*Lachnospira Roseburia Oscillospira Dialister*

**Table 3 biomedicines-11-00952-t003:** The impacts of certain drugs on the taxonomy of gut microbiota.

Drug Class	Substance	Increase	Decrease
Hypoglycaemic drugs	metformin	*Akkermansia muciniphila, Butyrivibrio, Bifidobacterium bifidum, Megasphaera, Escherichia coli, Shigella* spp. *Klebsiella, Citrobacter, Enterobacter cloacae, Salmonella enterica, Prevotella, Megasphaera,*	*Intestinibacter*, *Clostridium* spp., *Clostridium bartlettii*, *Mobiluncus mulieris, Peptoniphilus duerdenii, Eubacterium siraeum, Oscillospira, Barnesiellaceae Actinobacteria, Bacillus*
NSAIDs	acetylsalicylic acid	*Prevotella*, *Bacteroides, Barnesiella, Ruminococcaceae*	
	celecoxib, ibuprofen	*Acidaminococcaceae, Enterobacteriaceae Propionibacteriaceae*, *Pseudomonadaceae*, *Puniceicoccaceae, Rikenellaceae, Enterococcaceae* and *Erysipelotrichaceae*	
	celecoxib	*Enterococcaceae* and *Erysipelotrichaceae*	*Firmicutes, Ruminococcus*
PPIs		*Fusobacterium, Leptotrichia, Neisseria, Veillonella, Firmicutes, Streptococcaceae*, *Prevotellaceae*, *Campylobacteraceae, Enterococcus*, *Streptococcus*, *Staphylococcus*	*Bacteroidetes, Proteobacteria*
Statins	Atorvastatin, rosuvastatin	*Bifidobacterium longum* subsp. *longum*, *Anaerostipes hadrus* and *Ruminococcus obeum Bacteroides*, *Butyricimonas* and *Mucispirillum, Holdemanella, Faecalibacterium*	*Akkermansia muciniphila, Lactobacillus*
	fluvastatin	*Escherichia/Shigella*, *Ruminococcaceae UCG 014, Sutterella*	
Laxatives	15% polyethylene glycol (PEG)	*C. difficile*	

## Data Availability

Not applicable.

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
