# Peer review of "Clinical Relevance of Gut Microbiota Alterations under the Influence of Selected Drugs—Updated Review"

_biomedicines, 2023, doi:10.3390/biomedicines11030952_

Round 1

Reviewer 1 Report

Authors provided a comprehensive review concerning the roles of 6 classes of selected drugs and chemicals affecting microbiota homeostasis as it relates to therapeutic efficacy. A few small comments for authors to consider in the revision of the current version of the manuscript. 

In many locations of the manuscript words were separated by “-“ (e.g. at line 65 the word extent was spelled as ex-tent), probably caused by inappropriate line breaking in a previous draft of the manuscript. Authors should make an effort to remove all unnecessary “-“ throughout the entire manuscript.

As a review article, the “Materials and Methods” and “Results” section specifications seemed to be unnecessary and could mislead readers. Each of the six classes of drugs/chemicals listed under “Results” section could be listed as an independent section instead.  

Title for 3.6 should be reduced to “Phytochemicals” to be consistent with titles specified for 3.1 to 3.5.

The review seemed to be discussing previous studies concerning microbiota changes in the gut. If that is the case, the review title should be changed from “microbiota alterations” to “gut microbiota alterations”. If any referred work discussed microbiota changes in tissues/organs other than the gut, such as skin, authors should point that out specifically.   

Summary provided in Table 1 for phytochemicals is very good. Authors should make an effort to provide similar summary Tables for other chemicals/drugs if all possible to reduce redundancy in the description of details of the refereed studies.   

Author Response

RESPONSES TO THE REVIEWERS’ COMMENTS:

We would like to thank the Reviewers for their valuable comments, which allowed us to improve our manuscript. Below are the detailed responses and actions taken on the comments of the Reviewers.

Authors provided a comprehensive review concerning the roles of 6 classes of selected drugs and chemicals affecting microbiota homeostasis as it relates to therapeutic efficacy. A few small comments for authors to consider in the revision of the current version of the manuscript. 

In many locations of the manuscript words were separated by “-“ (e.g. at line 65 the word extent was spelled as ex-tent), probably caused by inappropriate line breaking in a previous draft of the manuscript. Authors should make an effort to remove all unnecessary “-“ throughout the entire manuscript..

Reply: We sincerely apologize for this. All “-“ used unnecessarily will be removed

As a review article, the “Materials and Methods” and “Results” section specifications seemed to be unnecessary and could mislead readers. Each of the six classes of drugs/chemicals listed under “Results” section could be listed as an independent section instead.  

Reply: The structure of the manuscript will be amended as suggested.

Title for 3.6 should be reduced to “Phytochemicals” to be consistent with titles specified for 3.1 to 3.5.

Reply: The name will be amended as suggested. Thank you.

The review seemed to be discussing previous studies concerning microbiota changes in the gut. If that is the case, the review title should be changed from “microbiota alterations” to “gut microbiota alterations”. If any referred work discussed microbiota changes in tissues/organs other than the gut, such as skin, authors should point that out specifically.   

Reply: we will specify in the amended title that the text refers to gut microbiota alterations only.

Summary provided in Table 1 for phytochemicals is very good. Authors should make an effort to provide similar summary Tables for other chemicals/drugs if all possible to reduce redundancy in the description of details of the refereed studies.  

Reply: Thank you for this suggestion. A table summarizing the paper content was added.

Reviewer 2 Report

In this review, the authors address a current topic which is the involvement of drugs in the modification of gut microbiota. The review is well written and includes interesting information

I have a few comments to the authors.

 1-  You said in the introduction that you are going to talk about the interactions between drugs and the gut microbiota. However, the rest of the manuscript deals almost only with the effect of drugs on the microbiota but not with the effect of a different composition of the gut microbiota on the efficacy and pharmacokinetics of drugs. Also, it would be interesting to discuss the drug interactions that are mediated by the alteration of the gut microbiota.

2- In the Materials and Methods section, you said that you have organized the data into 8 groups of drugs. However, in the results section, there are only 6 groups.

3- Figure 1 seems a bit complicated to read. I would ask the authors to better organize the figure and to add an explanatory legend.

4- In the statins section, you said on the one hand that these drugs increase the incidence of diabetes, and on the other hand that transplantation of oral microbiota using fecal material collected from rosuvastatin-treated mice improves hyperglycemia. Could you better explain this point?

5- In the same section, could you better explain the antibacterial effect of statins? For example, are the minimum inhibitory concentrations or minimum bactericidal concentrations lower than the plasma concentrations of these drugs in the clinic? and against which bacterial species?

Author Response

We would like to thank the Reviewers for their valuable comments, which allowed us to improve our manuscript. Below are the detailed responses and actions taken on the comments of the Reviewer.

In this review, the authors address a current topic which is the involvement of drugs in the modification of gut microbiota. The review is well written and includes interesting information

I have a few comments to the authors.

You said in the introduction that you are going to talk about the interactions between drugs and the gut microbiota. However, the rest of the manuscript deals almost only with the effect of drugs on the microbiota but not with the effect of a different composition of the gut microbiota on the efficacy and pharmacokinetics of drugs. Also, it would be interesting to discuss the drug interactions that are mediated by the alteration of the gut microbiota.

Reply: Thank you for this comment. We added a short paragraph on the two-face interaction, but more clearly underlined in the text that the article will cover information predominantly on the impact of drugs on microbiota structure.

In the Materials and Methods section, you said that you have organized the data into 8 groups of drugs. However, in the results section, there are only 6 groups.

Reply: we apologize for the mistake. The number was amended accordingly.

Figure 1 seems a bit complicated to read. I would ask the authors to better organize the figure and to add an explanatory legend.

Reply: a legend was added as requested.

In the statins section, you said on the one hand that these drugs increase the incidence of diabetes, and on the other hand that transplantation of oral microbiota using fecal material collected from rosuvastatin-treated mice improves hyperglycemia. Could you better explain this point?

Reply: We added a paragraph better explaining this issue, that still is under the research with conflicting data.

In the same section, could you better explain the antibacterial effect of statins? For example, are the minimum inhibitory concentrations or minimum bactericidal concentrations lower than the plasma concentrations of these drugs in the clinic? and against which bacterial species?

Reply: a paragraph was added as requested, better explaining this issue.